# Machine learning accurately predicts the multivariate performance phenotype from morphology in lizards

**Simon P. Lailvaux**[1]*, **Avdesh Mishra**[2], **Pooja Pun**[3], **Md Wasi Ul Kabir**[3], **Robbie S. Wilson**[4], **Anthony Herrel**[5], **Md Tamjidul Hoque**[3]*

1 Department of Biological Sciences, The University of New Orleans, New Orleans, LA, United States of America, 2 Department of Electrical Engineering and Computer Science, Texas A&M University-Kingsville, Kingsville, TX, United States of America, 3 Department of Computer Science, The University of New Orleans, New Orleans, LA, United States of America, 4 School of Biological Sciences, The University of Queensland, St. Lucia, Queensland, Australia, 5 Département Adaptations du Vivant, UMR 7179 C.N.R.S/M.N.H.N., Paris, France

* slailvaux@gmail.com (SPL); thoque@uno.edu (MTH)

**Data Availability Statement:** All model code and data, as well as a working software version of the MVPpred tool, are freely available and can be

## Abstract

Completing the genotype-to-phenotype map requires rigorous measurement of the entire multivariate organismal phenotype. However, phenotyping on a large scale is not feasible for many kinds of traits, resulting in missing data that can also cause problems for comparative analyses and the assessment of evolutionary trends across species. Measuring the multivariate performance phenotype is especially logistically challenging, and our ability to predict several performance traits from a given morphology is consequently poor. We developed a machine learning model to accurately estimate multivariate performance data from morphology alone by training it on a dataset containing performance and morphology data from 68 lizard species. Our final, stacked model predicts missing performance data accurately at the level of the individual from simple morphological measures. This model performed exceptionally well, even for performance traits that were missing values for >90% of the sampled individuals. Furthermore, incorporating phylogeny did not improve model fit, indicating that the phenotypic data alone preserved sufficient information to predict the performance based on morphological information. This approach can both significantly increase our understanding of performance evolution and act as a bridge to incorporate performance into future work on phenomics.

## Introduction

A major goal of evolutionary biology is accurate prediction of the phenotype from the genotype. The emerging field of phenomics in particular aims to quantify every aspect of the phenotype of an organism–that is, every measurable trait–and ultimately to relate it back, through several intermediate levels of biological organization, to the genome itself [1, 2]. However, while our ability to sequence genomes has advanced enormously in recent years, our capacity to characterize entire phenomes has not kept pace, particularly for phenotypes that are time

accessed at https://doi.org/10.6084/m9.figshare.18029474.v1.

**Funding:** This project was supported by the University of New Orleans, Office of Research in the form of an internal Interdisciplinary Grant Development Award to SL and MTH (CON 2946). The funders had no role in the study design, data collection, and analysis, decision to publish, or preparation of the manuscript.

**Competing interests:** The authors have declared that no competing interests exist.

consuming or otherwise difficult to quantify. Because some phenotypes are easier to measure than others, certain types of traits are either entirely absent from existing phenomes, or are described only in the most general terms [3]. Prime among these are those traits that describe how organisms conduct dynamic, ecologically relevant tasks such as jumping, running, flying, or biting, referred to collectively as whole-organism performance traits [4, 5].

Performance traits are key predictors of both survival and reproductive success in animals and as such form a cornerstone of the study of adaptation [4–6]. Performance is typically studied within the context of the ecomorphological paradigm, a statistical framework which states that morphology determines performance, which in turn affects fitness [7]. This paradigm has guided performance research for nearly 40 years and has been successfully applied to understand variation in morphology, performance, and fitness in a variety of animal species and over multiple levels of biological organization [8]. However, properly measuring maximum performance is time consuming, and doing so for suites of multiple performance traits in the same animals has proven to be a significant challenge. Consequently, despite intense interest in performance over the last several decades [6, 9–11], the entire whole-organism performance phenotype, comprising all or even most of the performance abilities of which a given species is capable, has therefore seldom been rigorously quantified [12]. Furthermore, even in cases where animals within a sample can be measured for multiple performance traits, the resulting datasets are rarely comprehensive, usually being limited to only two or three performance traits, and are typically characterized by missing data [e.g. 13]. Individual datapoints might fail to be collected for reasons ranging from logistical constraints and equipment failure to lack of cooperation of the subject being measured or even lack of continued availability of a given subject or species. These missing individual-level data cause further problems at the population and species levels for the analysis of evolutionary trends in particular. For example, comparative analyses of multiple phenotypic traits across a phylogeny are sensitive to missing data because even a single absent data point (i.e., mean value) for a given trait can force the exclusion of an entire species, reducing the overall power of the analysis. Approaches to incomplete comparative datasets based on imputing "placeholder" values, such as the PHYLOPARS method, do allow for the execution of an analysis that would not otherwise run with missing trait values [14, 15], but the accuracy of these methods is likely to be variable, frequently unverifiable, and prone to error at worst, particularly for situations with large amounts of missing data, or where missing data are not dispersed randomly across taxa.

One approach to addressing these issues is to predict data rather than measure it. The deterministic relationship between morphology and performance in particular offers scope for the prediction of unmeasured performance from individual morphology [16]. However, despite both the utility of the ecomorphological paradigm and the clear general validity of the morphology-to-performance relationship, modeling performance as a function of morphology alone is not always straightforward. Performance expression can be moderated, enhanced, or constrained by a variety of factors, including behavior [17]; energetic costs [14, 18–20]; elastic storage mechanisms [21, 22]; and the often complex relationships among performance and other facets of the integrated organismal phenotype [23–26]. Such constraints are especially relevant when animals are required to conduct multiple, yet different performance tasks on a day-to-day basis, many of which have conflicting morphological bases that cannot be optimized simultaneously. This can lead to trade-offs among specific performance traits such that specialization for one trait precludes high levels of expression in another [27, 28]. For example, birds such as gannets that dive from great heights to capture prey up to 30m below the water surface are often poor fliers because the ideal requirements for deep diving (high mass) and flying (low mass) are opposite [29]. Although intuitive, similar trade-offs among suites of several performance traits have proven difficult to uncover due in part to individual variation in

performance expression [13, 30, 31]. The existence of many-to-one mapping, whereby the same performance trait is produced by different morphological forms [32, 33], is a further complication for accurately predicting whole-organism performance. Consequently, the extension of this predictive scenario to a multivariate morphology-performance situation involving numerous, potentially conflicting performance traits is yet more challenging. Collectively, these constraints significantly limit our current ability to accurately predict multiple performance traits from a given underlying morphology.

The requirement for large-scale performance phenotyping coupled with existing constraints on both the measurement and prediction of multivariate performance demands that we adopt a new perspective on either performance measurement or inference. In the present study, we develop a machine learning method to accurately predict the multivariate performance phenotype from incomplete morphological datasets. Machine learning approaches are increasingly used at the whole-organism level to identify and analyze patterns within extremely large and detailed datasets often collected on only a handful of individuals. For example, machine learning techniques are used to extract meaningful biological signals from "noisy" patterns of individual movements recorded over long time periods using GPS trackers [34, 35], and to connect behavioral phenotypes to genetic sequences in populations of laboratory mice [2]. Furthermore, these methods can also be used to fill in "gaps" in large, complex datasets by deriving appropriate decision-boundaries for extrapolation, and ultimately to produce accurate predictions from novel data [36].

Here we adopt the latter approach, using machine learning to build an application to predict unmeasured maximum performance values at the level of the individual animal from a large and fragmentary dataset on lizard morphology and performance drawn from 68 species representing 8 different lizard families. Lizards are model organisms for the study of performance in general, and locomotion in particular [37]. We therefore take advantage of the substantial existing data on various lizard morphologies and associated performance phenotypes to train, test, and validate an machine learning model for imputing the multivariate performance phenotype from existing data on morphology. Specifically, we built a "stacked" machine learning model combining the outputs of several distinct regressor layers into a best-performing model that accurately predicts 5 distinct performance traits, including one complex, multicomponent trait (jumping ability) from 14 simple morphological measures across a range of diverse lizard taxa. Furthermore, we show that the addition of phylogenetic information on the relatedness of taxa in the model does not enhance model performance.

## Materials and methods

We built our machine learning model (hereafter termed MVPpred: "Multivariate Performance Phenotype Predictor") in three steps: missing value imputation; feature selection and classification; and stacking. Below we describe the nature of the training dataset, and outline briefly the process of model development and validation.

### Morphology and performance dataset

We assembled a training dataset comprising morphology and maximum performance data for nearly 2,000 individual lizards from 68 species. Data were sourced from the authors' personal datasets, contributions from other lizard performance researchers, and from publicly available data [38]. Performance data collected by different individuals and research groups are likely to be comparable given that whole-organism performance has the benefit of having standard protocols for maximum performance measurement to ensure that maximum values are recorded for each trait [39]. Morphology data are also commonly collected in a standardized manner,

and here comprise measurements of head dimensions (head length, head width, and head height); body size (snout-vent length [SVL] and body mass); individual limb elements (femur, tibia, metatarsus, longest hind toe, humerus, radius, metacarpal, longest fore toe); and tail length.

We considered 5 commonly-measured maximum performance traits that capture an array of diverse lizard terrestrial performance capacities: sprint speed (shortest time to traverse a set distance on a runway set at 45° or less to the horizontal because some lizard species tend to hop on horizontal surfaces [40]); endurance (longest time an individual is able to keep pace at a set, sub-$VO_2$ max speed on a treadmill before becoming exhausted [41, 42]); climbing (shortest time to traverse a set distance on a vertical runway [43, 44]); stamina (longest distance an individual is able to run when chased at maximum speed around a circular racetrack before becoming exhausted [45, 46]); jumping, (which is a composite variable comprising maximum distance, acceleration, velocity, and power of a jump at a given angle measured via a force plate or high-speed camera [47, 48]); and bite force (maximum force measured when a lizard is induced to bite down in a standardized manner on bite plates connected to a force transducer [49, 50]). However, because data were collected by different groups to test a variety of hypotheses, these data are, for many species, incomplete in terms of either the measured morphology or performance, or both. Furthermore, variation in the availability of data means that the machine learning training dataset is highly unbalanced in terms of both taxonomic representation and the amount of data available for each taxon (Fig 1); in particular, lizards of the genus *Anolis* are overrepresented relative to non-anoline lizards (see S23 Table for exact sample sizes in S1 File). Any extrapolations or inferences of performance->morphology relationships from such a sparse and fragmentary dataset using standard prediction methods such as model 1 or 2 regression are likely to be highly inaccurate; however, these data represent an ideal test case for machine learning approaches, as well as being representative of real-world data that are available to functional morphologists.

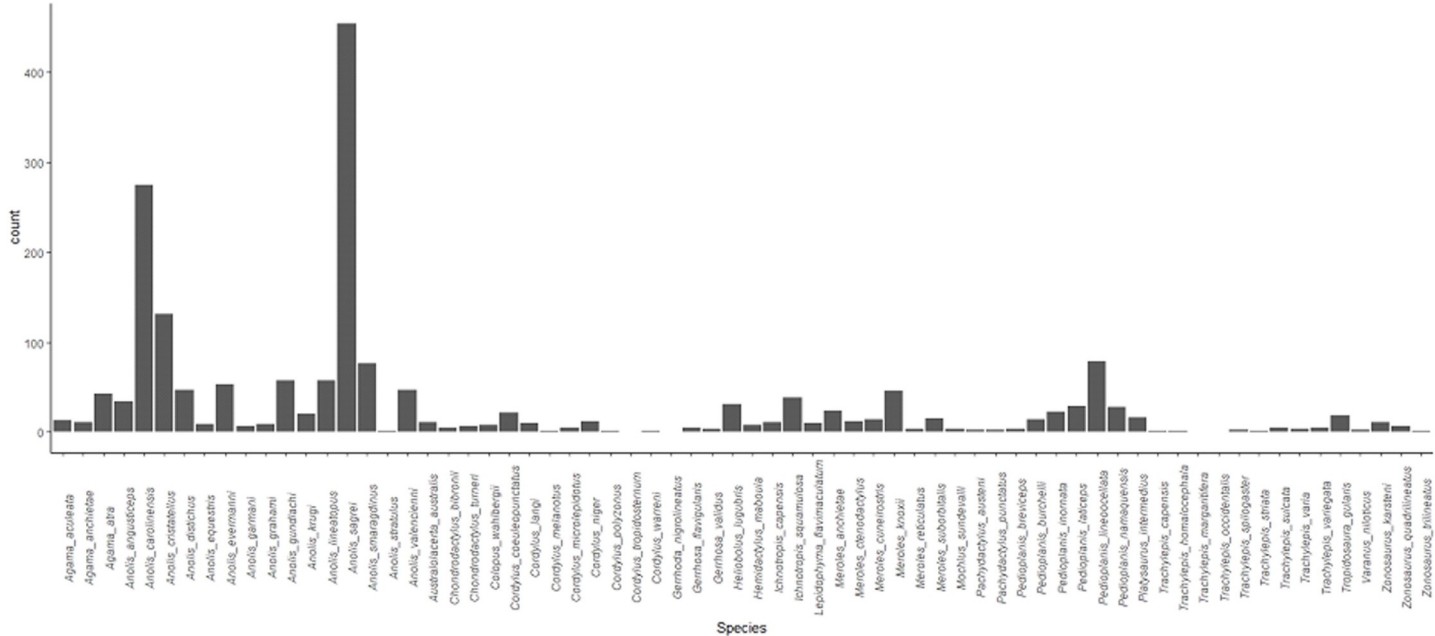

**Fig 1. Species names and sample size for each of the 68 taxa comprising the training and verification dataset.**

### Phylogenetic relationships

Comparative datasets comprising traits measured on multiple taxa must take into account the evolutionary relationships among those taxa because related species share an evolutionary history and thus are not independent data points [51]. Moreover, shared ancestry provides information that could be used to estimate missing data as phylogenetically closely related species will resemble each other in terms of both morphology and function. We pruned the large squamate phylogeny of Pyron et al. [52] to include only the species used in the current dataset (Fig 2). The evolutionary relationships among species were included in the base machine learning model as a distance matrix. However, this inclusion did not improve the predictions (see section A, "Two-step process" in the S1 File). Therefore, our final model is not affected by phylogeny; rather we used only the available morphology->performance dataset for training and confirmed the prediction accuracy by cross-validation.

### Handling missing values

We used the K-Nearest Neighbor (KNN) method to replace missing performance trait values in two steps. In the KNN method, the Euclidean distance between a target sample (S)

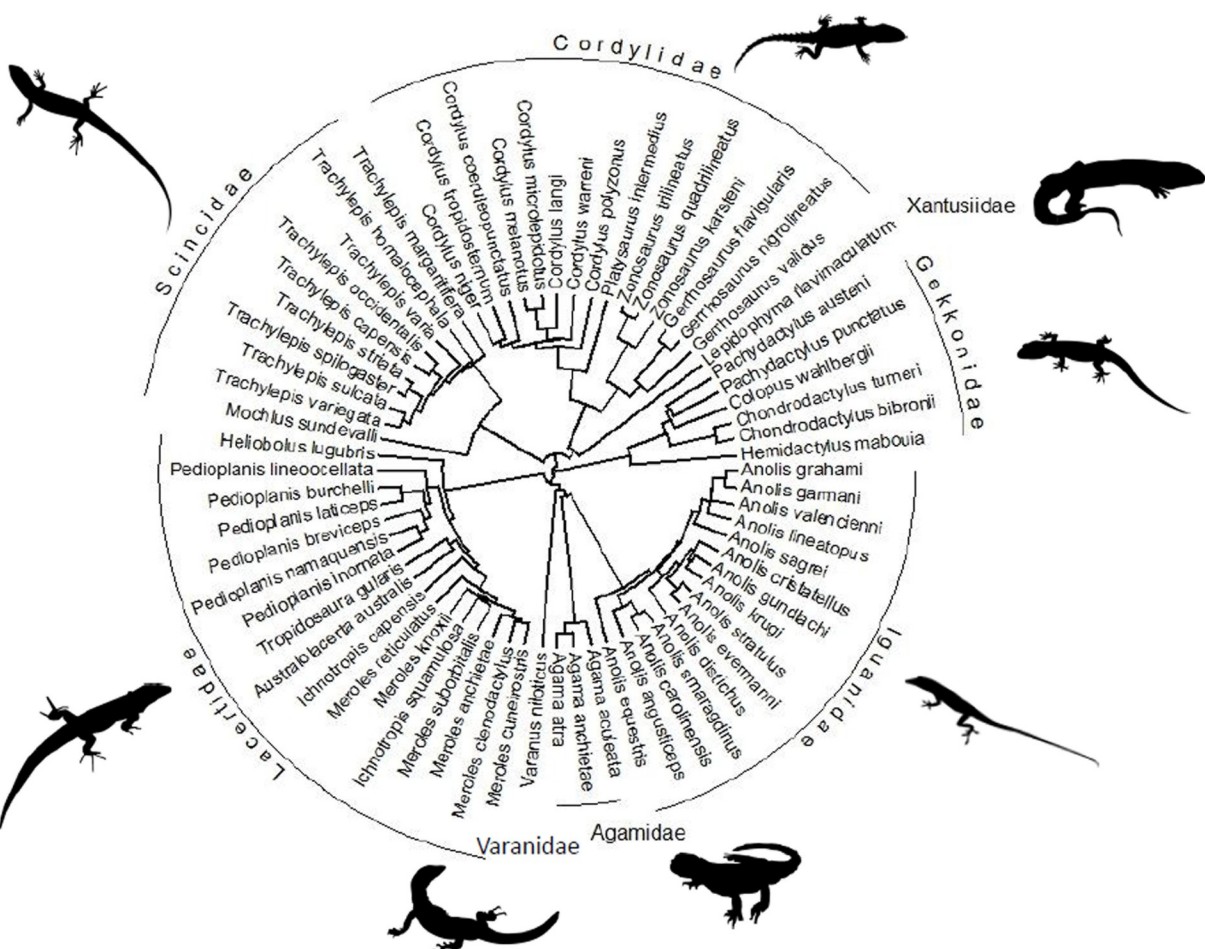

**Fig 2. Phylogenetic relationships among the 68 lizard taxa from 8 families included in the final model.** Note that phylogeny had no effect on the predictive accuracy of the final, stacked model.

**Table 1. Derivation of indices used to evaluate model classification and prediction.**

| Name of Metric | Definition |
|---|---|
| P | True value of the performance feature |
| Pavg | Mean of true values |
| Ppred | Predicted value of the corresponding performance feature |
| Ppred_avg | Mean of predicted values |
| N | Number of samples |
| Pearson Correlation Coefficient (PCC) | $\dfrac{\sum (P-Pavg)(Ppred-Ppred\_avg)}{\sqrt{\sum (P-Pavg)^2 \sum (Ppred-Ppred\_avg)^2}}$ |
| Mean Absolute Error (MAE) | $\frac{1}{N}\sum_{i=0}^{N-1}\lvert P - Ppred\rvert$ |
| Root Mean Square Error (RMSE) | $\sqrt{\frac{1}{N}\sum_{i=0}^{N-1}(P - Ppred)^2}$ |

belonging to a target feature (F; in this case, a performance phenotype or variable of interest) and all other samples is calculated. The missing value for S is then replaced by the average value of the K closest samples (i.e., those K with the lowest Euclidean distance). Initially, we applied the KNN method within each taxon using K = 5; however, since the dataset contained taxa with no values at all for certain features, we ultimately applied the method to the entire dataset. We found performance trait-specific different values of K while searching values from 3 to 200, and selected the appropriate value of K based on root mean square error (RMSE; see Table 1) for each performance feature (Table 2; the complete search outcome is presented in S1 File).

## Model performance evaluation

Our model consists of both a classification framework that predicts the taxon of a given sample and a regression framework that predicts a given sample's performance capacity. We measured the performance of the overall model using standard 10-fold cross-validation, whereby the data are divided into 10 sets of samples, 9 of which are used to train the prediction model while the remaining set is used to test the prediction model. We evaluated model performance using the Pearson Correlation Coefficient (PCC) and Mean Absolute Error (MAE) for the regression component (Table 1).

**Table 2. Optimum K-value search result (range 1 to 100), for various performance traits.**

| Feature | Optimum K (based on RMSE) | Root Mean Square Error (RMSE) |
|---|---|---|
| Jump power | 165 | 7.47 |
| Jump acceleration | 29 | 1.98 |
| Bite force | 57 | 4.53 |
| Jump velocity | 16 | 0.07 |
| Endurance | 154 | 32.08 |
| Sprint speed | 84 | 0.65 |
| Jump distance | 46 | 0.05 |
| Stamina | 25 | 3.66 |
| Angle | 34 | 1.84 |

## Regression framework

We used the available entire dataset to both train and validate our regression model. For standard training, we used 10-fold cross-validation (10 FCV), whereby we shuffled the dataset and divided it into ten sub-datasets by sequentially selecting equal individual samples at random without replacement.

We then evaluated the performance of the cross-validation using the Extra Tree Regressor (ETR) [53]; Gradient Boosting Regressor (GBR) [54]; Random Forest Regressor (RFR) [55], XGBoost Regressor (XGBR) [56], and Support Vector Regressor (SVR) [57].

i. Extra Trees Regressor (ETR): We have constructed the ET model with 1,000 trees, and the quality of a split is measured by the Gini impurity index.

ii. Random Forest Regressor (RFR): we have used a bootstrapping approach to construct 1,000 trees in the forest.

iii. XGBoost Regressor (XGBR): In our configuration of XGBR, the values of parameters: max_depth, eta, n_estimators, min_child_weight, subsample, scale_pos_weight, tree_-method, and max_bin are set to 6, 0.1, 100, 5, 0.9, 3, hist and 500 respectively and the rest of the parameters were set to their default value.

iv. Support Vector Regressor (SVR): For SVR, the RBF kernel parameter, γ, and the cost parameter, C are optimized to achieve the best 10-fold cross-validation accuracy using a grid search.

## Stacking framework

We further enhanced the performance of MVPpred using the stacking technique [58]. Briefly, the "no free lunch" theorem states that no single machine learning algorithm is best suited to all scenarios and datasets due to the associated generalization error [58–60] because one machine learning method would learn certain information from the dataset, whereas another would learn something different, depending on the specific underlying statistical learning principle. Stacking is an ensemble technique that combines information from multiple predictive models to generate a new model, and generally improves the prediction results through minimization of generalization error [61–63]. Here, the results (the difference between the predicted value and the original value) of different regressors used in the base layer along with the dataset provided to train the base layer are passed as a training dataset for the regressor used in the stacked meta layer. We explored different combinations (see Table 3) of base and meta layer.

# Results

## Outcome of the regression framework

Of the tested performance features, we found that jump power yields the best PCC and MAE (see Table 1 for the metric and Table 4 for the outcome) using R2 and optimized

**Table 3. Configurations of the five stacked models.**

|  | Base Layer | Meta Layer |
|---|---|---|
| SM1 | XGBR, RFR, GBR, ETR | ETR |
| SM2 | XGBR, RFR, GBR, ETR | GBR |
| SM3 | XGBR, RFR, GBR, ETR | RFR |
| SM4 | XGBR, RFR, GBR, ETR | XGBR |
| SM5 | XGBR, RFR, GBR, ETR | SVR |

**Table 4. Pearson correlation coefficient (PCC) and mean absolute error (MAE) of features.** Jump acceleration exhibited the highest prediction accuracy (bolded). To aid in the interpretation of MAE, we have also provided the mean value for each performance feature from the overall training dataset, as well as the associated standard errors. Note that MAE has the same units as the associated performance trait.

| Feature | Regression method | Mean (±SE) | PCC | MAE |
|---|---|---|---|---|
| Jump power (W/kg) | SVR | 45.94(±0.15) | 0.77 | 1.21 |
| **Jump acceleration** (m/s$^2$) | **SVR** | 32.17(±0.05) | **0.97** | **0.36** |
| Bite force (N) | GBR | 7.74(±0.18) | 0.94 | 1.35 |
| Jump velocity (m/s) | XGBR | 1.57(±0.002) | 0.95 | 0.02 |
| Endurance (s) | GBR | 213.71(±0.65) | 0.28 | 6.70 |
| Sprint (m/s) | RFR | 1.35(±0.02) | 0.88 | 0.23 |
| Jump distance (m) | ETR | 0.33(±0.001) | 0.84 | 0.01 |
| Stamina (m) | XGBR | 16.53(±0.11) | 0.83 | 1.42 |
| Angle | XGBR | 36.44(±0.06) | 0.75 | 0.527 |

Support Vector Regressor with RBF-kernel. From Table 4, we can see that jump acceleration was best predicted using the optimized SVR with RBF-kernel. The PCC (defined as a measure of linear correlation between the predicted and the actual value–see Table 1) is 0.97, and MAE (defined as the absolute difference between the predicted and the actual value) is 0.36 m/s$^2$. The results for jump acceleration using different regression methods are given in S7 Table of S1 File.

## Outcome of the stacking framework

We chose regressors for the base layer and meta-layers of the five stacked models (SM1, SM2, SM3, SM4, and SM5) based on Table 4. We used XGBR and SVR in the base layer of all stacking models because they exhibited the best PCC and MAE. The results from different stacking models for different features are summarized in Table 5 –however, the detailed results are available in S5 to S21 Tables of S1 File. SM2 outperformed the other three stacking models in all cases. PCC for these models ranged from 0.93 for jump distance to 0.99 for bite force, jump acceleration, and jump velocity, whereas MAEs were as low as 0.003m for jump distance (with a mean jump distance in the dataset of 0.33m), and as high as 1.73m for endurance (with a mean endurance value in the dataset of 213.71m) (Tables 4 and 5). Because of this superior performance, we used the SM2 stacking model throughout.

**Table 5. Pearson correlation coefficient (PCC) and mean absolute error (MAE) of different stacking models for various performance features.**

| Performance feature | Stacked configuration | PCC | MAE |
|---|---|---|---|
| Jump power (W/kg) | SM2 | 0.98 | 0.49 |
| Jump acceleration (m/s$^2$) | SM2 | 0.99 | 0.17 |
| Bite force (N) | SM2 | 0.99 | 0.57 |
| Jump velocity (m/s) | SM2 | 0.99 | 0.01 |
| Endurance (s) | SM2 | 0.95 | 1.73 |
| Sprint speed (m/s) | SM2 | 0.98 | 0.11 |
| Jump distance (m) | SM2 | 0.93 | 0.003 |
| Stamina (m) | SM2 | 0.98 | 0.63 |
| Angle | SM2 | 0.97 | 0.20 |
| | Average | **0.973** | **0.434** |

### Final software

To predict a given performance feature, the final software uses the prediction of the other eight performance features along with the morphological features as input. Our model is highly accurate even in the absence of phylogenetic information describing the relatedness among species. The final, stacked MVPpred model allows researchers to enter simple and easily obtained morphological data for an individual lizard and obtain accurate predictions for each of the 9 performance features pertaining to that individual. Furthermore, researchers could conceivably do this for all individuals in a sample, yielding a population or species mean for each trait that could then be used in comparative analyses. The results are available in S22 Table of S1 File.

## Discussion

Measuring every phenotype of a given organism on the scale that phenomics demands may not be possible, necessitating a demand for imputed or inferred data to at least some extent [2]. This will require both a paradigm shift in how we view data that are inferred but not measured from real organisms, and an accompanying advancement in the methods that we use to do so. We built, trained, and validated a machine learning model, which we call MVPpred, to accurately estimate unmeasured maximum performance data from a large dataset on lizard morphology and performance at the level of the individual animal. Our final stacked models predicted maximum multivariate performance with high accuracy, and cross-validation of our approach shows that the final, stacked MVPpred model significantly outperforms both simple statistical prediction methods such as ordinary least squares regression and single machine learning prediction methods in all cases. The prediction accuracy in terms of PCC of the stacked models ranged from 0.93 to 0.99, with low MAE in all cases (ranging from 0.003 to 1.73). Overall, our model was able to generate accurate predictions, even for performance traits that were poorly represented in the training dataset.

In addition to imputing the most likely maximum values of relatively simple performance metrics such as sprint speed or bite force, we also successfully and accurately predicted a more complex performance capacity. Jumping ability is itself a multivariate performance trait that can be characterized in several different ways [16, 64]. Some researchers have assessed individual jumping ability through relatively simple metrics such as maximum jump distance [65], whereas others have focused on describing both the kinetics and kinematics of jumping ability through measurement not only of distance, but also the velocity, acceleration, power, and the take-off angle of a jump [64, 66], all of which are interrelated and can trade-off against each other to shape overall jump trajectories [47, 48]. Our model predicted missing data for five key aspects of maximum jump performance (power, distance, acceleration, velocity, and angle), and did so with > 95% accuracy in all cases, suggesting that these methods hold the potential to predict other complex performance traits in different taxa as well.

Machine learning has been used to understand performance in the past. In particular, sports scientists have previously applied similar methods to the performance of individual athletes and events [36]; for example, Maszczyk et al. [67] used neural networks to predict the distance of javelin throws, and a similar approach was applied by Edelmann-Nusser et al. [68] to the women's 200m backstroke. Our study extends this approach to non-human animals in two key ways. First, our model predicts multiple maximum performance abilities as opposed to only one, including the five components of one complex performance ability (i.e., jumping). Second, we do this across 68 different species from 8 different families of lizards comprising a diversity of morphologies and ecological contexts. Our dataset was necessarily opportunistic and consequently is highly unbalanced with regard to species representation, ranging from

species represented by several hundred individuals (*Anolis carolinensis*), to others represented by only a handful of lizards (e.g., *Cordylus melanotus;* see Fig 1). Such datasets are typically not ideal for comparative studies aimed at identifying interspecific patterns [69, 70], making it all the more remarkable that our model was able to make accurate predictions even for sparsely sampled taxa. This combined multivariate and multispecies application will allow researchers to predict not only individual maximum performance for the traits of interest, but also for multiple traits across multiple species, granting increased flexibility in cases where missing performance data that cannot otherwise be obtained might compromise phenomic or comparative analyses.

Although accurately predicting maximum performance variables relating to existing data is valuable in itself, our model goes further and also opens up potential new avenues of investigation. MVPpred produces accurate predictions even in the absence of a known phylogeny, hinting at the potential universality of form-function relationships that might be obscured by variation at different levels of biological organization [see [31, 71], and [13] for examples at the within-species level]. However, although the aim of the current paper was to produce a model that accurately predicts multiple performance capacities, and although we validated those predictions against real data, the MVPpred model in its current form offers little insight into the causality underlying several of the predicted morphology->performance relationships. For example, while traits such as sprint speed and the various jump performance variables have clear deterministic relationships between limb morphology and the magnitude of the performance phenotype that are based on simple mechanical principles (e.g. Bauwens and Garland [72]), relationships between morphology and endurance are less clear cut. Endurance capacity is a function of oxygen delivery and cardiovascular function, which are not directly reflected in simple limb dimensions, and distribution of mass across the organism is more important than mass itself in determining endurance capacity [73]. The biomechanical basis of our model to accurately predict endurance from these morphological data is therefore not immediately apparent, and likely stems from the ability of the model to compute and compare not only relationships between predictors and predicted variables, but relationships among predicted performance variables as well. An important next step is therefore to interrogate our model to uncover and understand these causal relationships as well as any latent predictors that might exist. As such, models such as MVPpred also offer the possibility of a more complete understanding of form-function relationships at the whole-organism level as well and, potentially, a new approach for testing and understanding such relationships. Yet another possibility presented by our model performance, particularly in its accuracy in predicting performance for novel morphologies, is that an expanded and appropriately trained version of MVPpred could in principle allow for the accurate prediction of performance abilities from the bones of extinct organisms that have no living analogues. Similarly, our model could potentially represent a foundation for expanding this predictive approach to encompass other taxa and modes of locomotion beyond terrestrial lizards.

The accurate prediction of unmeasured data is a potentially valuable approach, but it also comes with some necessary caveats. Firstly, MVPpred predicts only maximum performance capacities. Although our focus on maximum performance here is consistent with much of the whole-organism performance literature, animals do not always perform to their maximum limits in nature [74], and there are many situations where it might be more useful or appropriate to use all of the available performance data, not just the maximum values, or to explicitly consider submaximal values [13, 75]. Second, despite both the power and generalizability of machine learning approaches and the lack of influence of phylogeny on our results, our model has only been formally validated with data from the 68 species represented in the training dataset (see Fig 1 and S23 Table for the full species list in S1 File). This model should therefore be

applied to individuals from other lizard species with caution, if at all. Expansion of the MVPpred model to encompass other species could be achieved by incorporating morphology and performance data pertaining to those species of interest.

In conclusion, MVPpred predicts multiple different whole-organism performance traits, including aspects of a multivariate performance trait (jumping ability) with a high degree of accuracy from even sparsely sampled data. Although we do not believe that this approach either is or should become a replacement for rigorous collection of real data where such collection is feasible, our model is nonetheless a clear improvement on existing imputation methods for missing performance data. The ability to accurately impute missing data across species is likely to enable further progress in integrating whole-organism performance and phenomics; understanding variation in form-function relationships; and ultimately in inferring unmeasured performance traits from novel morphologies.

## Supporting information

**S1 File. This word file presents the results of species-wise cross validation using the best stacking model, wherein we test a given species' performance by training the model with data from other species.** These results demonstrate good cross-species predictions where adequate training and testing data are available, suggesting that the model is useful even in the absence of phylogenetic information.
(DOCX)

## Acknowledgments

Thanks to J.F. Husak, A.M. Cespedes, and J.B. Losos for generously supplying additional morphology/performance data, and to J.F. Husak and H.E. Hanson for valuable comments on earlier drafts of the ms.

## Author Contributions

**Conceptualization:** Simon P. Lailvaux, Robbie S. Wilson.

**Data curation:** Simon P. Lailvaux, Md Wasi Ul Kabir, Anthony Herrel.

**Formal analysis:** Avdesh Mishra, Pooja Pun, Md Wasi Ul Kabir, Md Tamjidul Hoque.

**Investigation:** Avdesh Mishra, Md Tamjidul Hoque.

**Methodology:** Simon P. Lailvaux, Robbie S. Wilson, Md Tamjidul Hoque.

**Project administration:** Simon P. Lailvaux, Md Tamjidul Hoque.

**Resources:** Pooja Pun, Anthony Herrel, Md Tamjidul Hoque.

**Software:** Pooja Pun, Md Tamjidul Hoque.

**Supervision:** Simon P. Lailvaux, Avdesh Mishra, Anthony Herrel, Md Tamjidul Hoque.

**Validation:** Avdesh Mishra, Pooja Pun, Md Wasi Ul Kabir.

**Writing – original draft:** Simon P. Lailvaux.

**Writing – review & editing:** Simon P. Lailvaux, Avdesh Mishra, Pooja Pun, Md Wasi Ul Kabir, Robbie S. Wilson, Anthony Herrel, Md Tamjidul Hoque.

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
