## [Decision Letter · Decision Letter 0]

9 Jun 2021

PONE-D-21-11098

Machine Learning Accurately Predicts the Multivariate Performance Phenotype from Morphology in Lizards

PLOS ONE

Dear Dr. Lailvaux,

Thank you for submitting your manuscript to PLOS ONE. After careful consideration, we feel that it has merit but does not fully meet PLOS ONE’s publication criteria as it currently stands. Therefore, we invite you to submit a revised version of the manuscript that addresses the points raised during the review process.

We look forward to receiving your revised manuscript.

Kind regards,

Christopher Nice, Ph.D.

Academic Editor

PLOS ONE

Journal Requirements:

Additional Editor Comments (if provided):

Due to unavailability of reviewers for this manuscript, I am providing my own review to augment reviewer 1's comments. In any revision, please address both sets of comments.

AE comments:

This new approach for imputing missing data is well-presented and should be of high interest to readers involved with phenomics. My only real complaints are 1) that more details of the data used to train the model be presented and 2) that the authors are too succinct - some expansion of ideas and statistical issues or limitations would add value to this manuscript. Below are some elaborations on this theme and minor comments.

146: why "MVPpred"? The authors might justify the name of their approach.

153: change semicolons to commas

195-203: this needs some clarification, I think, with respect to the value of K. Was K=5 the initial approach, but then, given extensive missing data, the range of K was expanded (3-200) and the appropriate K chosen on the basis of MAE? Or is it RMSE? The supplementary figures (using MAE) do not comport with table 1. As an arbitrary example, for sprint speed, K=4 based on MAE from Fig. S6 but is reported in Table 1 as 84.

Tables 3 and 4 should be switched in order - the stacking combinations (methods and currently table 4) should precede the results (table 3). "PCC", "MAE", "BAse Layer" and "Meta Layer" should be defined in the legends.

Presentation of results: overall, this seems overly succinct and these three short paragraphs do not do justice for the work the authors have done here. For example, in the regression results, jump power is reported because of highest predictive power, and jump acceleration is mentioned at the end (250) but what about overall conclusions? Range of results. In other words, it is difficult to comprehend what the authors' main point is here. In the next section on stacking results (257) do the authors mean that SM2 outperformed the others in ALL cases as indicated by Table 5? No, according to the supplementary tables, the text is correct and the table is mistaken. I think readers would benefit from a fuller exploration of the results.

Discussion: as above with the results, the discussion could be broadened to help readers appreciate the full scope. Yes, MVPpred should not be a replacement for collecting data, but, as the authors state in the introduction and the first sentence of the Discussion, such data is logistically difficult to come by, or impossible. This model seems to be a valuable tool as a consequence. But readers might appreciate a discussion of the details. How and why are stacking methods improving results? Why is the SM1 configuration superior in all (or most) features? Are there limitations (beyond inferring causality), or inappropriate uses of this approach?

Reviewers' comments:

Reviewer's Responses to Questions

**Comments to the Author**

1. Is the manuscript technically sound, and do the data support the conclusions?

Reviewer #1: No

2. Has the statistical analysis been performed appropriately and rigorously? 

Reviewer #1: No

3. Have the authors made all data underlying the findings in their manuscript fully available?

Reviewer #1: No

4. Is the manuscript presented in an intelligible fashion and written in standard English?

Reviewer #1: Yes

5. Review Comments to the Author

Reviewer #1: This manuscript describes a model for predicting performance of lizards from morphological measurements. It makes two claims: one is that the model is extremely accurate at predicting these performances, and the second is that this is related to phenomics: high-throughput, multivariate phenotyping. The first claim is not supported by the kind of evidence one needs to evaluate it. The second is a non-sequitur. If you don’t measure the phenotype, you are not phenotyping.

A method which successfully predicts performance would be an extremely useful tool, and this software pipeline may be such a tool. The problem here is that the two measures reported here do not adequately summarize accuracy. High R^2 is certainly a good thing, but it is not enough without seeing the distribution of the data and the predictions used to validate the model. To take a really simple example, if we have two clusters of jump performances, with one cluster being can’t jump, and the other similar in jumping ability, you can get a very high R^2 without the model being accurate: all it has to do is say one is low, and the other is high. The MAE values are essentially uninterpretable without knowing the mean performance and the units in which they are measured. The proudly reported value of MAE of 1.48 has NO units attached!!! Proportional error of the predictions would be a very meaningful statistic.

There are excellent biological reasons to doubt that the repeatabilities of these performance characteristics could be high enough to produce the R^2 values reported. If you run or jump the same lizard and different equations, I would be astonished if repeatabilities could ever approach 98%. The assay for sprint speed is descrbied as doen at an angle of 45 or less. How can it not matter whether you sprint uphill on on the flat? Since this is the case how could an individuals morphology ever predict performance with such high accuracy. In short, the manuscript makes an extraordinary claim, but fails to back it up with meaningful statistical evidence.

What is absolutely essential to evaluate the accuracy is that we see the number of species actually measured in each case, as well as the error of each prediction when used in the test data set. Similarly, we really need to know whether the within-species variation is also explained by morphology, or whether this is just the mean that is explained. Since there is no enumeration of studies, I don’t know whether they are predicting the performance of 10 species from that of 58, or predicting 58 from 10 for each performance trait. What are the within-species sample sizes? How are those 2000 individual lizards divided among each trait, and species? Why is there no figure showing measurements and predictions with meaningfull errors for each?

If this model is in fact highly accurate by some meaningful measurement, this would be a very important result. It has important implications for phenomics, suggesting the dimensionality of the phenotype as whole is not so very high. Rather than BEING phenomics, such predictive ability would suggest that we do not really need phenomics. Trying to present the results AS phenomics is misguided, but the implications FOR phenomics are very interesting.

6. PLOS authors have the option to publish the peer review history of their article (what does this mean?). If published, this will include your full peer review and any attached files.

Reviewer #1: **Yes: **David Houle

---

## [Author Response · Author response to Decision Letter 0]

14 Aug 2021

Response to AE

Major Reviews

1. 146: why "MVPpred"? The authors might justify the name of their approach.

Response:

It is standard in computer science to provide names for machine learning models. In this case, “MVPpred” is a short form for “Multivariate Performance Phenotype Predictor”.

2. 153: change semicolons to commas

Response: 

Changed as suggested. 

3. Tables 3 and 4 should be switched in order - the stacking combinations (methods and currently table 4) should precede the results (table 3). "PCC", "MAE", "BAse Layer" and "Meta Layer" should be defined in the legends.

Response: 

Thank you. This issue has been corrected. We also switched the order of tables 1 and 2 for similar reasons.

4. 195-203: this needs some clarification, I think, with respect to the value of K. Was K=5 the initial approach, but then, given extensive missing data, the range of K was expanded (3-200) and the appropriate K chosen on the basis of MAE? Or is it RMSE? The supplementary figures (using MAE) do not comport with table 1. As an arbitrary example, for sprint speed, K=4 based on MAE from Fig. S6 but is reported in Table 1 as 84.

Response:

This is correct. The initial choice of K was 5; later, as the Editor noted, due to the extensiveness of missing data, the search range for the value of K was expanded to 3-200, and the appropriate value of K was selected based on the RMSE for each feature. With regard to the supplementary figure, we erroneously placed the results from our older experiment, where we compared K with respect to MAE. In the revised supplementary document, we have corrected the error and place the correct figures that show the plot of K versus RMSE (please see revised Fig. S1-S8). Thank you for pointing this out. 

5. Presentation of results: overall, this seems overly succinct and these three short paragraphs do not do justice for the work the authors have done here. For example, in the regression results, jump power is reported because of highest predictive power, and jump acceleration is mentioned at the end (250) but what about overall conclusions? Range of results. In other words, it is difficult to comprehend what the authors’ main point is here. In the next section on stacking results (257) do the authors mean that SM2 outperformed the others in ALL cases as indicated by Table 5? No, according to the supplementary tables, the text is correct and the table is mistaken. I think readers would benefit from a fuller exploration of the results.

Response: 

We have expanded on the results as requested. We have supplied the correct supplement in this revision, which is in agreement with the results reported in the manuscript. The confusion regarding SM1/SM2 was a result of uploading an earlier, incorrect supplement – SM2 is indeed the best performing model in all cases. 

6. Discussion: as above with the results, the discussion could be broadened to help readers appreciate the full scope. Yes, MVPpred should not be a replacement for collecting data, but, as the authors state in the introduction and the first sentence of the Discussion, such data is logistically difficult to come by, or impossible. This model seems to be a valuable tool as a consequence. But readers might appreciate a discussion of the details. How and why are stacking methods improving results? Why is the SM1 configuration superior in all (or most) features? Are there limitations (beyond inferring causality), or inappropriate uses of this approach?

Response:

We have expanded upon the results in the revised version as requested. With regard to stacking improving results, traditional machine learning methods such as K-NN, Logistic Regression, Random Forest, SVM, etc., do not perform as well as the stacking method because of the associated generalization error. Specifically, one machine learning method would learn certain information from the dataset whereas, the other would learn different information. The type of information an individual method learns from the data is totally dependent on the specific statistical learning principle based on which it was designed. The SM2 (not SM1 – see above) model has the greatest predictive power in all cases, although it isn’t possible to say why it works better, just that it does.

The stacking approach allows us to layer algorithms that do one or more things particularly well on top of each other, allowing us to refine model outputs iteratively within each layer. Because the “no free lunch” theorem in computer science tells us that no single algorithm is best suited to all scenarios and datasets, choosing configurations that are better in some or most cases (in this case, algorithms that optimize several features better than others) is a necessity. 

As requested, we have also expanded upon the limitations to our approach in the revised manuscript.

Response to the Reviewer

Major Reviews

1. This manuscript describes a model for predicting performance of lizards from morphological measurements. It makes two claims: one is that the model is extremely accurate at predicting these performances, and the second is that this is related to phenomics: high-throughput, multivariate phenotyping. The first claim is not supported by the kind of evidence one needs to evaluate it. The second is a non-sequitur. If you don’t measure the phenotype, you are not phenotyping.

Response: 

We believe that a lack of clarity on our part in the original manuscript has led to some misunderstanding as to the accuracy of our model, based at least in part on confusion regarding the level at which our model operates and is validated. 

To be clear: our model interpolates missing data at the level of the individual animal for multiple different performance traits. Furthermore, our model is also tested and validated at the level of the individual, not the species level. Thus, researchers can enter into the MVPpred model whatever morphological measures they have for an individual animal, and the model will accurately impute the resulting 9 maximum performance capacities for that individual. The validation of the model happens at the level of the individual as well. We provide more detail on this procedure below and in the revised paper, which we hope the reviewer finds satisfactory. 

The same problem underlies the evident confusion regarding phenotyping. Houle et al. (2010) define phenomics as “the acquisition of high-dimensional phenotypic data on an organism-wide scale”. We believe our proposed approach is consistent both with this definition, and with the way in which machine learning has been used to bolster phenotyping in phenomics studies in the past, specifically in the application of this method at the level of the individual animal. Again, we address this point in more detail in our response to the final comment (5) below.

2. A method which successfully predicts performance would be an extremely useful tool, and this software pipeline may be such a tool. The problem here is that the two measures reported here do not adequately summarize accuracy. High R^2 is certainly a good thing, but it is not enough without seeing the distribution of the data and the predictions used to validate the model. To take a really simple example, if we have two clusters of jump performances, with one cluster being can’t jump, and the other similar in jumping ability, you can get a very high R^2 without the model being accurate: all it has to do is say one is low, and the other is high. The MAE values are essentially uninterpretable without knowing the mean performance and the units in which they are measured. The proudly reported value of MAE of 1.48 has NO units attached!!! Proportional error of the predictions would be a very meaningful statistic.

Response: 

Every individual in the dataset was used for both testing and training using K-fold Cross-validation. Note: We were careful to ensure that the same sample being tested was not included in the training of the model. The R2 values reported is defined as (1 – u/v), where u is the residual sum of square ((y_true – y_pred) ** 2).sum() and v is the total sum of squares ((y_true – y_true.mean()) ** 2).sum(). Therefore, R2 in general refers to the correlation between predicted and actual data using all individuals in the dataset. The value of R2 will only be high if and when the predicted values are very close to the actual value, which indicates that the model is highly accurate. We agree that if the imputed missing values are the same as the existing values present in the data sample, we would be introducing redundancy in the data, which could lead to a high R2 value. However, while applying K-nearest neighbor algorithm in our application of missing value imputation, we search for K-nearest neighbor in the entire dataset. Next, a unique value of K for individual morphological performance traits is identified by considering a large search space of 3 – 200. Then, the missing value is imputed taking the average of the values from only the nearest neighbor samples regardless of species type. These steps help us ensure that redundancy is not introduced in the dataset indicating that the R2 scores are not biased.

We have provided the average performance values and corresponding standard errors for each feature to facilitate interpretation of the MAEs. We also note that MAE here has units of the performance feature in question. With regard to the distribution of the data among performance traits, while this would be too much to summarize in a manuscript, we note that we are making all of the data freely available for anyone to inspect. Certainly, there are cases where performance data are sparse, but again the imputed data are tested and verified against all of the other data, including cases where those data are present.

3. There are excellent biological reasons to doubt that the repeatabilities of these performance characteristics could be high enough to produce the R^2 values reported. If you run or jump the same lizard and different equations, I would be astonished if repeatabilities could ever approach 98%. The assay for sprint speed is descrbied as doen at an angle of 45 or less. How can it not matter whether you sprint uphill on on the flat? Since this is the case how could an individuals morphology ever predict performance with such high accuracy. In short, the manuscript makes an extraordinary claim, but fails to back it up with meaningful statistical evidence.

Response:

 There seems to be some confusion here as well; the R2 values here refer to the accuracy of the model (i.e. predicted vs actual values; see previous response), NOT the repeatabilities of the empirical data. It is important to note that over the last 40+ years, performance researchers have almost always striven to measure maximum performance and have developed standardized protocols based on measuring a given performance trait multiple times and selecting the maximum value (see Losos et al. 2002 for a detailed description). Consequently, we focus on maximum performance here as well. Although previous studies have shown that such performance measures are in fact often highly repeatable, the extraordinary repeatabilities we report here are a result of the machine learning approach being applied to the entire dataset, not of empirical data collection. 

With regard to angled tracks sometimes being used to collect sprint data, many lizards, especially those in the genus Anolis, tend to hop on horizontal surfaces rather than run. Performance researchers therefore found that angling the track as described elicits proper sprinting. One might reasonably imagine that lizards sprinting up an angled track would be slower than those sprinting horizontally because sprinting at an angle requires more power, but this is not the case; lizard sprinting is not limited by power output, and sprint speeds do not decrease as angle increases (see Farley 1997; Irschick et al. 2003).

We have clarified the above issues in the revised manuscript.

4. What is absolutely essential to evaluate the accuracy is that we see the number of species actually measured in each case, as well as the error of each prediction when used in the test data set. Similarly, we really need to know whether the within-species variation is also explained by morphology, or whether this is just the mean that is explained. Since there is no enumeration of studies, I don’t know whether they are predicting the performance of 10 species from that of 58, or predicting 58 from 10 for each performance trait. What are the within-species sample sizes? How are those 2000 individual lizards divided among each trait, and species? Why is there no figure showing measurements and predictions with meaningfull errors for each?

Response: 

The reviewer is correct that information on species composition and sample sizes should be provided. We have included a figure in the main document (now Figure 1), and a table in the supplement (Table S23) giving both the names and the sample sizes for each of the 68 species in our training dataset.

We used K-fold cross-validation, whereby the dataset is randomly subdivided and then those subdivisions tested iteratively against the remaining data, on the entire dataset, making sure that the sample used in testing is not included in the training of the model. Because the individuals are selected for those K subdivisions at random, it isn’t the case that we are using x species to predict the performance of all other species; instead, we are selecting x INDIVIDUALS and testing their performance against that of all other individuals, and those individuals are randomly selected without regard for species identity. 

Not only are we not using a certain number of species to predict other species, but species identity is not used in the model at all; as we report elsewhere in the ms, neither species identity nor the phylogeny are informative for predicting performance data from morphology.

5. If this model is in fact highly accurate by some meaningful measurement, this would be a very important result. It has important implications for phenomics, suggesting the dimensionality of the phenotype as whole is not so very high. Rather than BEING phenomics, such predictive ability would suggest that we do not really need phenomics. Trying to present the results AS phenomics is misguided, but the implications FOR phenomics are very interesting.

Response: 

Again, we do not believe (and explicitly do not claim) that our approach is a replacement for empirical phenotyping; rather, we view it as a method for imputing such data when they cannot be acquired through more conventional means. This general procedure is already used in phenomics to fill in gaps on data acquired from individuals; our expanded approach enables us to do this for individuals belonging to multiple different species.

To be clear, our method does not simply generate a species mean that can be used for comparative or phylogenetic analyses; it allows us to generate missing datapoints at the level of the individual for 68 different lizard species. In this respect, it is entirely consistent with the way that machine learning has been applied to phenomics in the past, just expanded to 68 different species as opposed to only one. Furthermore, we disagree with the reviewer that this method, powerful as it is, either is or should be a replacement for empirical data collection, although we do think it has enormous utility as a supplementary approach. We have revised the manuscript to make this point clearer.

 

References

Farley, C.T. (1997) Maximum speed and mechanical power output in lizards. The Journal of Experimental Biology, 200, 2189-2195.

Irschick, D.J., Vanhooydonck, B., Herrel, A. & Andronescu, A. (2003) The effects of loading and size on maximum power output and gait characteristics in geckos. Journal of Experimental Biology, 206, 3923-3934.

Losos, J.B., Creer, D.A. & Schulte, J.A. (2002) Cautionary comments on the measurement of maximum locomotor capabilities. Journal of Zoology, 258, 57-61.

---

## [Decision Letter · Decision Letter 1]

30 Sep 2021

PONE-D-21-11098R1Machine Learning Accurately Predicts the Multivariate Performance Phenotype from Morphology in LizardsPLOS ONE

Dear Dr. Lailvaux,

Thank you for submitting your manuscript to PLOS ONE. After careful consideration, we feel that it has merit but does not fully meet PLOS ONE’s publication criteria as it currently stands. Therefore, we invite you to submit a revised version of the manuscript that addresses the points raised during the review process. The reviewers found this revision to be improved and more clear. However, Reviewer 1 in particular is interested in more details regarding model validation / performance evaluation. Given that clarity is at a premium with the introduction of new methods, I ask that you consider these comments carefully.

We look forward to receiving your revised manuscript.

Kind regards,

Christopher Nice, Ph.D.

Academic Editor

PLOS ONE

Journal Requirements:

Additional Editor Comments (if provided):

Reviewers' comments:

Reviewer's Responses to Questions

**Comments to the Author**

1. If the authors have adequately addressed your comments raised in a previous round of review and you feel that this manuscript is now acceptable for publication, you may indicate that here to bypass the “Comments to the Author” section, enter your conflict of interest statement in the “Confidential to Editor” section, and submit your "Accept" recommendation.

Reviewer #1: (No Response)

Reviewer #2: All comments have been addressed

Reviewer #3: (No Response)

2. Is the manuscript technically sound, and do the data support the conclusions?

Reviewer #1: Partly

Reviewer #2: Yes

Reviewer #3: Yes

3. Has the statistical analysis been performed appropriately and rigorously? 

Reviewer #1: No

Reviewer #2: I Don't Know

Reviewer #3: Yes

4. Have the authors made all data underlying the findings in their manuscript fully available?

Reviewer #1: No

Reviewer #2: Yes

Reviewer #3: Yes

5. Is the manuscript presented in an intelligible fashion and written in standard English?

Reviewer #1: Yes

Reviewer #2: Yes

Reviewer #3: Yes

6. Review Comments to the Author

Reviewer #1: The alterations to the manuscript having improved its clarity on many basic points. However, now that I understand what the authors are doing better, it is clear that some of my original questions remain. Tthe authors appear to have misread my comments on measures of model accuracy. Because the evaluation of the accuracy of the model predictions is dubious in many respects, I am still worried that the authors give quite a misleading picture of their model’s performance.

First, I want to be clear on a critical point, and that is whether the model is being asked to predict imputed data in the 10-fold cross-validation step. The authors describe imputation, and then fitting of the overall model to the imputed + observed data in some detail. However, the performance evaluation is simply described as splitting the data into subsets, followed by standard cross-validation. There are two potential issues here.

I would like to assume that the authors removed the imputed performance values from the 1/10 of the data that is used as the test set. Please make explicit that this is not what you are doing. Only actually observed maximum performance values are legitimate to use in the test data set evaluate the accuracy of the model predictions. While it is fine to use imputed values if they improve model predictions, it is NOT a test of accuracy to use a test data set with imputations to see whether the imputations are predicted by the model. Statements like this one at line 297-298 make me afraid that the authors HAVE made this fundamental error “treadmill endurance was measured for only ~7.8% of individuals but inferred with an accuracy of 0.95 throughout the dataset.” If they have, then all measures of accuracy in this manuscript are completely bogus.

The more general issue is that an overall measure of correlation or error is not very informative, particularly in a data set with massively unbalanced sample sizes. As the authors note the data set is heavily weighted to Anolis, and indeed 45% of the specimens are in just three species. This means that whatever individual-wise measure of accuracy is computed is mostly reflecting the model’s performance in those species with the largest sample sizes. For example authors say (lines 323-325) “making it all the more remarkable that our model was able to make accurate predictions even for sparsely sampled taxa.” The problem is that the authors have not calculated the accuracy of predictions on sparsely sampled taxa, just on the overall data set. To make such statements the authors need to report accuracy for each species. The authors single PCC or MAE value is effectively weighted by within species sample size. A more representative overall measure would be an unweighted mean (or median) PCC or MAE value over species.

This would help get at another unaddressed issue from my previous review and that is the issue of proportional errors. While it is a great improvement that we now have the overall means of each performance measure, an MAE value might be very small for an organism with a high predicted value, and very large for an organism with a small predicted value. There is still no summary of the actual performance values to enable a reader to evaluate this issue, as the authors make no attempt to do so. The statement on line 295 about MAE is still made without units, and is completely meaningless without thinking in proportional terms. For example the extreme values cited 0.003 meters jump and 1.75 seconds are each close to a 1% error, and not really very different. But is the error low throughout the range of performance values? Interpret all MAE values proportionally.

Another cross-validation that would also be informative is to do it species-wise – that is leave out each species from the training data set and ask how well its performance is predicted by the remaining species performance data. This would be an interesting test of the author’s contention that phylogeny does not matter much. If that is true then there should be little cost to cross-species predictions. If cross-species predictions do well within the data set, then this would suggest that the model may indeed be useful when applied to species whose performance has not been measured.

I still argue that prediction is not really that relevant to phenomics, except in the way outlined in my previous review. As the lead author on the article cited, perhaps you should give my point of view some credence.

Minor comments.

Line 105 Cormorants dive from the water surface, not while in flight.

Line 262-271. I believe the authors mean to refer readers to Tables S4 and S5 in this paragraph.

Reviewer #2: Dear Editor,

I have considered the revised manuscript entitled 'Machine Learning Accurately Predicts the Multivariate Performance Phenotype from morphology in Lizards’ by S. Lailvaux, although I did not review the original version. I also read carefully the authors’ extensive responses to the referees' comments and questions and paid special attention to how these comments were addressed (where necessary) in the manuscript (the added track-changes version was very helpful for this). In my opinion, the authors did this revision very thoroughly and respectfully.

With regard to the content, I must admit that I am not at all familiar with the statistical (and computer) techniques applied to this massive lizard-dataset. On the other hand, I am sufficiently familiar with ecological/morphological research (in the evolutionary context) to see the enormous potential of this methodology. This is especially the merit, also for the 'mathematical layman', of the very comprehensible introduction and discussion. The only thing I cannot quite assess is what the direct applicability (and thus in a sense the valorisation value) of this method might be to other, new cases (e.g. a study of the link between morphology and performance traits in arthropods). Morphometric and performance data will always be needed to train the routine to make predictions. But how extensive should this training dataset be? How 'lizard' specific is the current procedure (in other words: is the protocol directly applicable to other systems)? Can the classical 'ecologist' apply this method autonomously ... or will the participation of colleagues from computer sciences be necessary? Etc. The authors may wish to provide a perspective in this respect in a short paragraph in the discussion.

Reviewer #3: A concise report on the application of a statistical method to morphology>performance data as a way to deal with incomplete datasets. The paper addresses a real problem and provides a robust statistical solution. I've got no major criticisms of the methods or results. I have a few comments that might be considered to help improve, especially, the discussion:

line 120: the model only predicts performance for incomplete performance datasets, correct? Or can the model work with incomplete morphological datasets too? Please check the wording here.

line 144: ML. Every time I read ML I think maximum likelihood but it is machine learning. I might suggest avoiding this abbreviation altogether the paper already has a lot of abbreviations, so one fewer would make for a bit less mental work for the reader.

line 297: I am having a hard time understanding how this is even possible. So if I measure endurance on 7.8% of my samples and then use the model to predict the other 92.2% of samples...how do I 'know' if that prediction is at all accurate? You don't really 'know' what those endurance values are, you only have a prediction based on other traits that is dependent upon very poor samples of 'known' values. Do the authors really believe that if I measured the other 92.2% of species that my measurements would fall within the prediction 95% of the time (or rather does the model tell us that)??? It seems like a huge leap of faith based on very weak underlying sampling.

Line 341: I think you've missed a key idea. While endurance (i.e. the ability of muscles to sustain contraction to propel the animal forward at a given speed) is no doubt most closely linked to the cardiovascular and pulmonary systems....it is also TOTALLY dependent upon the limb morphology and body dimension of a given species. Shorter limbed species must cycle their limbs more often to maintain speed, thus taxing their muscles more than a longer limbed species that ran the same distance. Dachsunds will always tire before greyhounds and some (or even a lot) of that is related to their limb shape!

Line 366: Ughh...you undercut one of the main benefits of your model...prediction. But it is true that applying this model to other species is iffy. You might consider some text here to explain what type of dataset might be needed to build a model that ultimately COULD be used across other species.

Line 334: Feels like a bit of a cop out. I think there is more causality that you can infer here than you give yourself credit for. Or rather, there is more biology here than the paper currently digs into. I realize the point of the paper is to demonstrate and validate the statistical model....but it sure would have been nice to see a bit deeper dive into the biology of how these performance traits trade-off or facilitate, etc.

7. PLOS authors have the option to publish the peer review history of their article (what does this mean?). If published, this will include your full peer review and any attached files.

Reviewer #1: **Yes: **David Houle

Reviewer #2: No

Reviewer #3: No

---

## [Author Response · Author response to Decision Letter 1]

2 Dec 2021

Response to First Reviewer

The alterations to the manuscript having improved its clarity on many basic points. However, now that I understand what the authors are doing better, it is clear that some of my original questions remain. The authors appear to have misread my comments on measures of model accuracy. Because the evaluation of the accuracy of the model predictions is dubious in many respects, I am still worried that the authors give quite a misleading picture of their model’s performance.

First, I want to be clear on a critical point, and that is whether the model is being asked to predict imputed data in the 10-fold cross-validation step. The authors describe imputation, and then fitting of the overall model to the imputed + observed data in some detail. However, the performance evaluation is simply described as splitting the data into subsets, followed by standard cross-validation. There are two potential issues here.

I would like to assume that the authors removed the imputed performance values from the 1/10 of the data that is used as the test set. Please make explicit that this is not what you are doing. Only actually observed maximum performance values are legitimate to use in the test data set evaluate the accuracy of the model predictions. While it is fine to use imputed values if they improve model predictions, it is NOT a test of accuracy to use a test data set with imputations to see whether the imputations are predicted by the model. Statements like this one at line 297-298 make me afraid that the authors HAVE made this fundamental error “treadmill endurance was measured for only ~7.8% of individuals but inferred with an accuracy of 0.95 throughout the dataset.” If they have, then all measures of accuracy in this manuscript are completely bogus.

- We thank the reviewer for these comments; however, we strongly reject the characterization of our model validation methods as “dubious”. In particular, the author is incorrect to state that “only actually observed maximum performance values are legitimate to use in the test data set”.

- It is important here to be clear: none of the approaches that we use for model assessment and validation in this paper are in any way novel, unusual, or controversial within the field of machine learning, and we implement them here in standard ways. However, because we understand the potential for confusion, we lay out the rationale for our approach involving both the data imputation and model validation here in some detail. 

- The machine model learning occurs in two stages. First, the model learns how best to impute all the missing values via appropriate interpolation. However, this is not done by simply taking an “unintelligent” column-wise (i.e. trait- or feature-wise) mean (which we call k=N where N equals the total instances in the entire dataset), nor by choosing a single nearby “best” value (called k =1). Instead, we used the k-nearest neighbor (KNN) machine-learning approach, which outperforms both these and other simple statistical methods (see Jerez et al. 2010 for an example and validation). In KNN, the value of k indicates how many neighboring datapoints we use for interpolation and averaging. In our case, figures S1 to S8 in the manuscript supplement describe the search for the optimum value of k for each performance trait, which we found to be 165 for jump power (Fig. S1); 57 for bite force (Fig. S3), and so on. We are therefore not simply choosing a nearby datapoint to insert, nor are we always averaging the same number of nearby datapoints; rather, we search for the optimal value of nearby datapoints to average and choose that value of k based on the lowest observed error rate. So the imputation procedure itself is an important component of the model’s accuracy, and choosing appropriate averages to use for the imputation of each missing datapoint is the first stage at which the error variance of the imputed value prediction is reduced. 

- The second stage of machine learning involves using the entire dataset, including those imputed values, to train the model, using the various algorithms and alsorithm combinations that we describe in the manuscript. It is important that those imputed values are also used, because if we were to discard them the model would be left with very few datapoints from which to learn. So including the imputed values at this and later stages is a key part of the ability of machine learning to produce accurate predictions from sparse datasets – which is the entire purpose of machine learning. 

- Although the reviewer did not question the imputation procedure, we have included this explanation here to give context to the part of the methodology that the reviewer did question – namely the validation procedure, and the inclusion of those imputed data in the test dataset. To answer that question clearly: we did include the imputed data in the test datset, and again we did this because that is the standard methodology for validating models that make predictions from datasets comprising large amounts of missing data. Excluding the imputed data from the test dataset leaves one with a very sparse test dataset, such that many of the folds in the k-fold cross-validation are left with zero or only a very few datapoints on which to effectively test the model predictions (see new Excel sheet in supplementary data). So including the imputed data ensures that the performance is not artificially low, as would be the case if it were excluded. Furthermore, we have an estimate of the error involved in our model from the very start (i.e. during imputation see Figures S1-S8 again). Those error rates in computing the values are, with the exception of endurance, extremely low, and that error only decreases as the learning algorithms and stacking technique are applied. Consequently, inclusion of the imputed data in the test data does not introduce enormous amounts of error. Finally, it is of note that although we are of course able to evaluate the performance of the final model, the processes of model prediction and validation occur iteratively as part of the training/learning procedure (see Methods sections “Model Performance Evaluation” and “Regression Framework” in the ms), and as such removing imputed data from the model makes little sense. 

- The above notwithstanding, one can, of course, perform the cross-validation without the imputed data as the reviewer suggests, and also with varying size test folds. Unsurprisingly, test folds containing sparse data perform poorly, again because they contain insufficient datapoints to test the model predictions. However, this does leave us with a moderate number of folds with sufficient original data for testing, which yield consistent and very good results, further supporting our contentions here regarding the accuracy of the model. 

The more general issue is that an overall measure of correlation or error is not very informative, particularly in a data set with massively unbalanced sample sizes. As the authors note the data set is heavily weighted to Anolis, and indeed 45% of the specimens are in just three species. This means that whatever individual-wise measure of accuracy is computed is mostly reflecting the model’s performance in those species with the largest sample sizes. For example authors say (lines 323-325) “making it all the more remarkable that our model was able to make accurate predictions even for sparsely sampled taxa.” The problem is that the authors have not calculated the accuracy of predictions on sparsely sampled taxa, just on the overall data set. To make such statements the authors need to report accuracy for each species. The authors single PCC or MAE value is effectively weighted by within species sample size. A more representative overall measure would be an unweighted mean (or median) PCC or MAE value over species.

This would help get at another unaddressed issue from my previous review and that is the issue of proportional errors. While it is a great improvement that we now have the overall means of each performance measure, an MAE value might be very small for an organism with a high predicted value, and very large for an organism with a small predicted value. There is still no summary of the actual performance values to enable a reader to evaluate this issue, as the authors make no attempt to do so. The statement on line 295 about MAE is still made without units, and is completely meaningless without thinking in proportional terms. For example the extreme values cited 0.003 meters jump and 1.75 seconds are each close to a 1% error, and not really very different. But is the error low throughout the range of performance values? Interpret all MAE values proportionally.

-As the reviewer suggests, we have created an excel file to show the results of cross-species prediction. The number of test samples represents the number of samples we have for each species after removing the missing values. As the new table S24 shows, most of the species only have a small number of samples. It is not possible to train and test the model species-wise with such a small number of samples without conducting imputation first.

- We have provided the 10-fold cross-validation result with the range (Min and Max) of each performance feature to interpret all the MAE proportionally. These cross validation results are summarized in Table S34, and the error is relatively low considering the range of values for the performance features.

Another cross-validation that would also be informative is to do it species-wise – that is leave out each species from the training data set and ask how well its performance is predicted by the remaining species performance data. This would be an interesting test of the author’s contention that phylogeny does not matter much. If that is true then there should be little cost to cross-species predictions. If cross-species predictions do well within the data set, then this would suggest that the model may indeed be useful when applied to species whose performance has not been measured.

I still argue that prediction is not really that relevant to phenomics, except in the way outlined in my previous review. As the lead author on the article cited, perhaps you should give my point of view some credence.

- As the reviewer suggested, we have calculated species-wise cross-validation with the best stacking model. We test a species’ performance by training the model with other species data. The table is too big to display here, so we attach an excel file as a supplementary document to demonstrate the results. The prediction accuracy is far better for those with more extensive training and testing data. For example, for the performance feature Bite force (Values ranging between 0-109.26 and17.3% missing), the average MAE and PCC are 7.42 and 0.48, respectively. The result clearly suggests that it is possible to have cross-species predictions with a good number of train and test samples, which supports our contention that phylogeny is of little importance within this dataset.

- With regard to the issue of phenomics, we are not indifferent to the reviewers’ point of view; the problem from our perspective is that the reviewer simply asserts that our approach holds no relevance to phenomics, but does not explain why. As we noted in our response to the first review and in the manuscript, other phenomic studies have and do use machine learning prediction as a matter of course, albeit on a smaller scale than we propose here; as we see it, our approach is entirely compatible with these and with the definition proposed in Houle et al. (2010). Clearly the reviewer disagrees; but without being privy to the reasons why, we are at a loss as to how to address his concerns.

Line 105 Cormorants dive from the water surface, not while in flight.

- This should have referred to gannets, which do exhibit this behavior. We have corrected it in the text. 

Line 262-271. I believe the authors mean to refer readers to Tables S4 and S5 in this paragraph.

- The table reference in the manuscript is correct as it stands. 

 

Response to Second Reviewer

I have considered the revised manuscript entitled ‘Machine Learning Accurately Predicts the Multivariate Performance Phenotype from morphology in Lizards’ by S. Lailvaux, although I did not review the original version. I also read carefully the authors’ extensive responses to the referees’ comments and questions and paid special attention to how these comments were addressed (where necessary) in the manuscript (the added track-changes version was very helpful for this). In my opinion, the authors did this revision very thoroughly and respectfully.

With regard to the content, I must admit that I am not at all familiar with the statistical (and computer) techniques applied to this massive lizard-dataset. On the other hand, I am sufficiently familiar with ecological/morphological research (in the evolutionary context) to see the enormous potential of this methodology. This is especially the merit, also for the ‘mathematical layman’, of the very comprehensible introduction and discussion. The only thing I cannot quite assess is what the direct applicability (and thus in a sense the valorisation value) of this method might be to other, new cases (e.g. a study of the link between morphology and performance traits in arthropods). Morphometric and performance data will always be needed to train the routine to make predictions. But how extensive should this training dataset be? How ‘lizard’ specific is the current procedure (in other words: is the protocol directly applicable to other systems)? Can the classical ‘ecologist’ apply this method autonomously ... or will the participation of colleagues from computer sciences be necessary? Etc. The authors may wish to provide a perspective in this respect in a short paragraph in the discussion.

- Given that our model uses and makes predictions only on data from lizards, we believe that this model is only relevant to lizards. Indeed, we further adopt the conservative position that this model is best applied only to lizard species for which the model has been validated (i.e. those species included in the model), even though our model also suggests that phylogeny has no affect on model performance (see also response to reviewer 1 above). We do already address these points explicitly in the final two paragraphs of the discussion. Briefly, we view this model as a starting point beyond which we might expand the model to other taxa and modes of locomotion via inclusion of appropriate training data.

Response to Third Reviewer

line 120: the model only predicts performance for incomplete performance datasets, correct? Or can the model work with incomplete morphological datasets too? Please check the wording here.

- Our training dataset comprises both incomplete morphology and performance data, as we note in the methods (lines 171-173). The model is able to deal with both types of missing data because machine learning methods do not distinguish between dependent and independent variables, and rather use the entire dataset for prediction.

l

ine 144: ML. Every time I read ML I think maximum likelihood but it is machine learning. I might suggest avoiding this abbreviation altogether the paper already has a lot of abbreviations, so one fewer would make for a bit less mental work for the reader.

- We have replaced the abbreviation “ML” with “machine learning” throughout as suggested.

line 297: I am having a hard time understanding how this is even possible. So if I measure endurance on 7.8% of my samples and then use the model to predict the other 92.2% of samples...how do I ‘know’ if that prediction is at all accurate? You don’t really ‘know’ what those endurance values are, you only have a prediction based on other traits that is dependent upon very poor samples of ‘known’ values. Do the authors really believe that if I measured the other 92.2% of species that my measurements would fall within the prediction 95% of the time (or rather does the model tell us that)??? It seems like a huge leap of faith based on very weak underlying sampling

- We understand the reviewer’s skepticism. One of the strengths of machine learning is its ability to predict novel, unmeasured data from a sparse dataset, and that is the reason we use it here. We refer the reviewer to our response to reviewer 1, where we address in detail this concern and the methods by which machine learning imputes missing data. Nonetheless, we agree that it may be best to be conservative in our claims, and consequently we have removed this statement from the manuscript.

Line 341: I think you’ve missed a key idea. While endurance (i.e. the ability of muscles to sustain contraction to propel the animal forward at a given speed) is no doubt most closely linked to the cardiovascular and pulmonary systems....it is also TOTALLY dependent upon the limb morphology and body dimension of a given species. Shorter limbed species must cycle their limbs more often to maintain speed, thus taxing their muscles more than a longer limbed species that ran the same distance. Dachsunds will always tire before greyhounds and some (or even a lot) of that is related to their limb shape!

- We agree with the reviewer that even endurance performance is likely to depend on underlying morphology, consistent with the ecomorphological paradigm. However, we argue that simple limb dimensions, which this dataset comprises here, do not reflect that variation as they do for a trait such as sprint speed (where speed is stride length x stride frequency, and therefore clearly a function of limb length). 

- The reviewers’ point regarding interspecific variation is well taken; however, it is also important to note that that intraspecific endurance capacity can and does vary both among and within individuals in a way that, again, is not reflected in simple limb measurements, and that our dataset comprises this intraspecific variation in addition to interspecific variation. Indeed, much of the deterministic value of morphology in terms of endurance is dependent on the distribution of mass both along and among the limbs and other parts of the body, something that again is not captured by simple limb measurements, or even measures of whole body mass. We therefore stand by our original statement in the manuscript, although we have amended it to acknowledge the issue regarding mass distribution.

Line 366: Ughh...you undercut one of the main benefits of your model...prediction. But it is true that applying this model to other species is iffy. You might consider some text here to explain what type of dataset might be needed to build a model that ultimately COULD be used across other species.

- We address this point explicitly in the discussion, where we state that: “… Yet another possibility presented by our model performance, particularly in its accuracy in predicting performance for novel morphologies, is that an expanded and appropriately trained version of MVPpred could in principle allow for the accurate prediction of performance abilities from the bones of extinct organisms that have no living analogues. Similarly, our model could potentially represent a foundation for expanding this predictive approach to encompass other taxa and modes of locomotion beyond terrestrial lizards.” Our aim is, in fact, to do this as the next step once this proof-of-concept study is accepted. We have inserted an additional statement to this effect.

Line 334: Feels like a bit of a cop out. I think there is more causality that you can infer here than you give yourself credit for. Or rather, there is more biology here than the paper currently digs into. I realize the point of the paper is to demonstrate and validate the statistical model....but it sure would have been nice to see a bit deeper dive into the biology of how these performance traits trade-off or facilitate, etc.

- We agree with the reviewer that there is a lot more biology to be discussed. However, we think that to do so in this manuscript is to put the cart before the horse. Again, in our view, this manuscript represents a proof-of-concept of the model, and there is a great deal of scope to follow up this paper with the types of studies the reviewer suggests. 

References: 

Jerez, M.J., I. Molina, P.J. García-Laencina, E. Alba, N. Ribelles, M. Martin and L. Franco. 2010. Missing data imputation using statistical and machine learning methods in a real breast cancer problem. Artificial Intelligence in Medicine 50: 105-115.

---

## [Editor Report · Decision Letter 2]

7 Dec 2021

Machine Learning Accurately Predicts the Multivariate Performance Phenotype from Morphology in Lizards

PONE-D-21-11098R2

Dear Dr. Lailvaux,

We’re pleased to inform you that your manuscript has been judged scientifically suitable for publication and will be formally accepted for publication once it meets all outstanding technical requirements.

Kind regards,

Christopher Nice, Ph.D.

Academic Editor

PLOS ONE
---

## [Editor Report · Acceptance letter]

12 Jan 2022

PONE-D-21-11098R2 

Machine Learning Accurately Predicts the Multivariate Performance Phenotype from Morphology in Lizards 

Dear Dr. Lailvaux:

I'm pleased to inform you that your manuscript has been deemed suitable for publication in PLOS ONE. Congratulations! Your manuscript is now with our production department. 

Kind regards, 

on behalf of

Dr. Christopher Nice 

Academic Editor

PLOS ONE